# DNA Methylation and Gene Expression in Blood and Adipose Tissue of Adult Offspring of Women with Diabetes in Pregnancy—A Validation Study of DNA Methylation Changes Identified in Adolescent Offspring

**DOI:** 10.3390/biomedicines10061244

**Published:** 2022-05-26

**Authors:** Eleonora Manitta, Irene Carolina Fontes Marques, Sandra Stokholm Bredgaard, Louise Kelstrup, Azadeh Houshmand-Oeregaard, Tine Dalsgaard Clausen, Louise Groth Grunnet, Elisabeth Reinhardt Mathiesen, Louise Torp Dalgaard, Romain Barrès, Allan Arthur Vaag, Peter Damm, Line Hjort

**Affiliations:** 1Novo Nordisk Foundation Center for Basic Metabolic Research, Metabolic Epigenetics Group, Faculty of Health and Medical Sciences, University of Copenhagen, 2200 Copenhagen, Denmark; manitta@sund.ku.dk (E.M.); barres@sund.ku.dk (R.B.); 2Department of Obstetrics, Center for Pregnant Women with Diabetes, Rigshospitalet, 2100 Copenhagen, Denmark; irene.marques@campus.ul.pt (I.C.F.M.); louise.kelstrup@regionh.dk (L.K.); dr.houshy@gmail.com (A.H.-O.); pdamm@dadlnet.dk (P.D.); 3Department of Science and Environment, Roskilde University, 4000 Roskilde, Denmark; sandrabredgaard@hotmail.com (S.S.B.); ltd@ruc.dk (L.T.D.); 4Department of Clinical Medicine, Faculty of Health and Medical Sciences, University of Copenhagen, 2200 Copenhagen, Denmark; tine.clausen@regionh.dk (T.D.C.); elisabeth.reinhardt.mathiesen@regionh.dk (E.R.M.); 5Department of Obstetrics and Gynecology, Herlev and Gentofte Hospital, 2730 Herlev, Denmark; 6Novo Nordisk A/S, Novo Allé 1, 2880 Bagsværd, Denmark; 7Department of Obstetrics and Gynecology, Hillerød Hospital, 3400 Hillerød, Denmark; 8Steno Diabetes Center Copenhagen, Herlev Hospital, 2730 Herlev, Denmark; louise.groth.grunnet.02@regionh.dk (L.G.G.); allan.arthur.vaag@regionh.dk (A.A.V.); 9Department of Endocrinology, Rigshospitalet, 2100 Copenhagen, Denmark

**Keywords:** developmental programming, intrauterine hyperglycemia, gestational diabetes, type 1 diabetes, adipose tissue, *ESM1*, *MS4A3*, *TSPAN14*, DNA methylation, gene expression, epigenetics

## Abstract

Maternal gestational diabetes and obesity are associated with adverse outcomes in offspring, including increased risk of diabetes and cardiovascular diseases. Previously, we identified a lower DNA methylation degree at genomic sites near the genes *ESM1*, *MS4A3*, and *TSPAN14* in the blood cells of adolescent offspring exposed to gestational diabetes and/or maternal obesity in utero. In the present study, we aimed to investigate if altered methylation and expression of these genes were detectable in blood, as well in the metabolically relevant subcutaneous adipose tissue, in a separate cohort of adult offspring exposed to gestational diabetes and obesity (O-GDM) or type 1 diabetes (O-T1D) in utero, compared with the offspring of women from the background population (O-BP). We did not replicate the findings of lower methylation of *ESM1*, *MS4A3*, and *TSPAN14* in blood from adults, either in O-GDM or O-T1D. In contrast, in adipose tissue of O-T1D, we found higher *MS4A3* DNA methylation, which will require further validation. The adipose tissue *ESM1* expression was lower in O-GDM compared to O-BP, which in turn was not associated with maternal pre-pregnancy BMI nor the offspring’s own adiposity. Adipose tissue *TSPAN14* expression was slightly lower in O-GDM compared with O-BP, but also positively associated with maternal pre-pregnancy BMI, as well as offspring’s own adiposity and HbA1c levels. In conclusion, the lower DNA methylation in blood from adolescent offspring exposed to GDM could not be confirmed in the present cohort of adult offspring, potentially due to methylation remodeling with increased aging. In offspring adipose tissue, *ESM1* expression was associated with maternal GDM, and *TSPAN14* expression was associated with both maternal GDM, as well as pre-pregnancy BMI. These altered expression patterns are potentially relevant to the concept of developmental programming of cardiometabolic diseases and require further studies.

## 1. Introduction

Overnutrition and hyperglycemia in pregnancy have serious health implications for both the woman and the fetus, including complicated pregnancies, and can necessitate intensive neonatal care [1,2,3]. In addition, the children of women with diabetes in pregnancy (gestational diabetes mellitus (GDM) and type 1 diabetes mellitus (T1D)) have increased risk of long-term complications in adult life [4,5]. This includes an eightfold higher risk of pre-diabetes and type 2 diabetes (T2D) in GDM offspring (O-GDM) and a fourfold higher risk in T1D offspring (O-T1D) [5,6]. Despite evidence of an association between maternal diabetes and obesity in pregnancy and offspring metabolic complications deriving from epidemiological and observational studies [7,8], the molecular mechanisms underlying this association are poorly understood. 

Several studies have investigated the potential link between diabetes and obesity in pregnancy and epigenetic changes in the offspring, which might partly contribute to the aforementioned phenotypes [9,10]. The majority of these, however, only examined epigenetic differences in blood. Previously, we conducted an epigenome-wide association study in a GDM sub-cohort of the Danish National Birth Cohort and identified significant differences in blood DNA methylation between 9–16-year-old offspring of women with GDM versus offspring of control women (offspring from the background population, O-BP) [11]. Differential methylation between GDM and control offspring was observed at multiple genes, including a lower methylation degree at *ESM1* (Endothelial cell specific molecule 1), at *MS4A3* (Membrane spanning 4-domains A3), and at *TSPAN14* (Tetraspanin 14). When we examined the confounding effect of maternal pre-pregnancy BMI, we showed that lower methylation of *ESM1* associated independently with both maternal GDM and obesity, lower methylation of *MS4A3* associated only with maternal pre-pregnancy BMI, and lower methylation of *TSPAN14* associated only with GDM.

*ESM1* encodes a secreted protein, endocan, which is primarily produced by endothelial cells in lung and kidney tissue [12]. Endocan is upregulated under inflammatory conditions, and in cardiovascular disorders [13]. *MS4A3* gene product is primarily expressed in myeloid cells and has been suggested as a marker for early myeloid differentiation in human hematopoiesis [14], but has not previously been suggested to be involved in metabolism or cardiometabolic health. *TSPAN14* encodes a transmembrane protein regulating the activity of ADAM10 and NOTCH signaling [15,16], which have been implicated in several processes, including inflammation and atherosclerosis development [17,18]. Based on our previous findings of their lower DNA methylation degree in GDM offspring, and due to their potential roles in inflammatory and cardiovascular health, these genes may be relevant in influencing offspring phenotypes following in utero exposure to obesity and/or diabetes. In the present study, we therefore aimed to measure the DNA methylation and gene expression degree of the selected target genes in a separate cohort of adult offspring (median 30.5 years old) exposed to different degrees of glycemia and obesity in pregnancy (GDM, T1D, and controls). The adult cohort has previously been described in detail [19], and it is worth noting that the O-GDM, exposed to both maternal diabetes and obesity, were not more obese than the O-BP, but mainly differed in having an increased diastolic blood pressure and higher 2 h glucose levels post an oral glucose tolerance test (OGTT) compared to O-BP. 

We first aimed to test whether lower DNA methylation at *ESM1*, *TSPAN14*, and *MS4A3*, previously detected in blood from adolescent GDM offspring, was present in this older offspring cohort, and subsequently whether these methylation changes were also detectable in the metabolically important subcutaneous adipose tissue (SAT). The adipose tissue is particularly interesting, since chronic inflammation has been linked with two of the three target genes, and inflammation in adipose tissue is considered a crucial risk factor for later development of insulin resistance in overweight individuals [20]. Second, we investigated differences in the expression levels of the target genes in blood and SAT, and examined whether associations were present between the expression levels and selected clinical parameters. Finally, we aimed to investigate whether maternal hyperglycemia or maternal obesity in pregnancy was the primary driver of potential gene methylation and expression differences in the offspring.

## 2. Materials and Methods

### 2.1. Study Population

A follow-up study of a cohort of adult offspring (average age 31 years) of women with diabetes during pregnancy was conducted in 2012–2014, as previously described in detail [19]. The follow-up cohort comprised 206 adult offspring, including offspring of women with diabetes in pregnancy (*n* = 82 O-GDM and *n* = 67 O-T1D) and women from the background population (BP) (*n* = 57 O-BP). All the births were singleton pregnancies between 1978–1985 at the Obstetrics Department at Rigshospitalet, Copenhagen, Denmark. 

In Denmark, at the time the cohort was established, GDM was diagnosed when at least two of seven measurements obtained during a 3h 50 g oral glucose tolerance test (OGTT) exceeded the mean +3 standard deviations (SD) for a reference group of normal-weight non-pregnant women without a family history of diabetes [21,22]. All women with GDM in this study cohort were diet-treated only. As previously published, women with T1D were selected according to the following criteria: onset of diabetes at age ≤40 years, a classical history of hyperglycemic symptoms before disease diagnosis, and insulin treatment starting ≤6 months after diagnosis [5]. Women from the background population were from the local community and had been routinely referred for antenatal care and delivery at the same hospital. Baseline information regarding the health status of the women during pregnancy and the offspring in the perinatal period was accessible from the maternal medical records of the Department of Obstetrics at Rigshospitalet, Copenhagen. 

The cohort segmentation was based on empirical factors of the offspring’s exposure to intrauterine hyperglycemia and their respective expected genetic susceptibility to T2D. Therefore, O-GDM and O-T1D were both exposed to intrauterine hyperglycemia but represent distinct genetic predispositions to T2D. Whereas O-BP, were not exposed to intrauterine hyperglycemia and were estimated to have a relatively low genetic susceptibility to T2D (Figure 1). 

Offspring diagnosed with T1D, MODY (maturity onset diabetes of the young), other severe chronic diseases, or who were pregnant were excluded from participation. 

### 2.2. Clinical Examination and Tissue Collection

Abdominal SAT biopsy samples were collected after overnight fasting. The samples were collected through a small skin incision with a Bergström needle, immediately frozen in liquid nitrogen, and stored at −80 °C until further use. After tissue collection, a fasting venous blood sample was drawn; EDTA, serum, and Paxgene tubes were collected; and a 75 g OGTT was performed. Plasma lipid profile (LDL, HDL, triglycerides, and total cholesterol), HbA1c, glucose, insulin, and C-peptide concentrations were assessed. All samples were analyzed in agreement with standard laboratory procedures, as described previously [19]. Other clinical measures included height, weight, blood pressure, and body composition assessed by dual energy X-ray absorptiometry (DEXA) whole-body scan (Lunar Prodigy Advance; GE Medical Systems, Fairfield, CT, USA). Pregnancy and birth data, such as maternal pre-pregnancy BMI and OGTT, and offspring birthweight were available from the original medical records.

### 2.3. Candidate Gene Selection

The selection of the genes for the present study was based on the findings from our discovery study [11], with the primary aim of studying three candidates associated to maternal GDM and/or maternal obesity (*ESM1*, *MS4A3*, and *TSPAN14*). The genes were selected based on the highest effect size in the discovery study, after assessing their expression in the target tissues (blood and SAT) on the GTEx platform [23].

### 2.4. DNA Methylation Analysis

In total, 206 blood (buffy coat) samples and 138 SAT biopsies were available for DNA extraction. Genomic DNA was extracted from blood using a QIAamp 96 DNA blood kit, and from SAT using a QIAamp DNA Mini Kit (Qiagen, Germantown, MD, USA). Bisulfite conversion was performed using an Epitect Bisulfite Kit (Qiagen, Germantown, MD, USA). We studied DNA methylation at *ESM1*, *TSPAN14*, and *MS4A3* CpG sites in the promoter or gene body regions, as shown in Figure 2. The assays where designed to include the specific CpGs previously identified as being differentially methylated in adolescents exposed to maternal GDM and/or obesity (*ESM1*: cg00992687 and cg09452568, *MS4A3*: cg14328641 and *TSPAN14:* cg11411705) [11], and also allowed additional adjacent CpGs to be measured within the same assay. Measurement of DNA methylation degree was performed using the pyrosequencing technique (PyroMark Q48 Autoprep, Qiagen, Germantown, MD, USA) with PyroMark Q48 reagents. PCR and sequencing primers were designed using PyroMark Assay Design 2.0 (Appendix A). 

### 2.5. Gene Expression Analysis

A PAXgene Blood miRNA kit was used for total RNA extraction from blood, and a miRNeasy Mini Kit for total RNA extraction from SAT (both Qiagen, Germantown, MD, USA). Genomic DNA was removed from total RNA by DNase I treatment during RNA extraction. cDNA synthesis was performed with a QuantiTect Reverse Transcription Kit (Qiagen). Primer design for qPCR was carried out in the NCBI Primer-BLAST tool [24]. The selected primers are shown in Appendix A. mRNA expression of *ESM1*, *MS4A3*, and *TSPAN14*, and the reference gene *TFIIB* was measured using quantitative real-time PCR (q-PCR), using SYBR green reagents and a ViiA 7 Real-Time PCR System instrument. *TFIIB* was chosen as reference gene based on its known capacity for not changing with adipogenesis nor with altered glucose levels [25,26]. Despite *MS4A3* expression being identified in the GTEx platform, the expression of *MS4A3* mRNA was non-detectable in both the blood and adipose tissue samples of the current cohort, and it was, therefore, excluded from the qPCR analyses.

### 2.6. Statistical Analyses

Statistical analysis was conducted with RStudio software (version 4.1.0) and SAS Enterprise Guide (version 7.1). Graphic illustrations were developed with GraphPad Prism (version 9.0). Maternal GDM and T1D were treated as the exposure variables during data analysis, while gene expression and DNA methylation were the primary outcomes.

Categorical variables are presented in the form of counts (percentage). Continuous variables are presented with mean (±SD) for parametric distributions and with median (interquartile range (IQR)) for non-parametric distributions. Clinical differences between groups (O-GDM vs. O-BP and O-T1D vs. O-BP) were evaluated via Student *t* test and Wilcoxon rank score for continuous variables with parametric and non-parametric variables, respectively, and via χ^2^ test for categorical variables. 

The association between DNA methylation or gene expression and the exposure variables of diabetes status and pre-gestational BMI (as a continuous measure) were tested using multivariate regression models, including both exposure variables in the same model, and also adjusting for offspring sex. Data assumptions for regression analysis were carried out by residual visualization in quantile-quantile plots (QQ plots) and the results are shown as estimated mean differences (β coefficient) and the 95% confidence interval (CI). 

Correlation analyses between gene expression and methylation level, and between gene expression and offspring clinical parameters were performed via Spearman rank analysis. 

All tests were two-tailed and *p* values <0.05 were considered significant.

## 3. Results

### 3.1. Clinical Characteristics

The clinical characteristics of the pregnant women are summarized in Appendix A. Glucose levels during pregnancy were obtained from women with T1D and GDM at different time points. The degree of intrauterine hyperglycemia was most likely of higher severity in women with overt T1D versus GDM, as only women with mild GDM (diet-treated) were included in the study. This assumption is further supported by the markedly higher rate of LGA infants among O-T1D. Clinical characteristics of the offspring cohort, including data from the pregnancies and birth outcomes are summarized in Table 1. As previously published, the mean maternal pre-gestational BMI was 3 kg/m^2^ higher in the O-GDM group, when compared to the control/background population offspring group (O-BP) (*p* < 0.001). Both groups exposed to diabetes in pregnancy presented with higher 2 h plasma glucose concentrations during the OGTT when compared to O-BP (*p* ≤ 0.001); however, the glucose levels of all three offspring groups were still within the normal range. Diastolic blood pressure was higher in O-GDM compared to O-BP (*p* = 0.02) (Table 1) [21].

### 3.2. DNA Methylation Validation

We analyzed DNA methylation at CpGs in the *ESM1*, *MS4A3*, and *TSPAN14* regions in both blood and SAT, comparing the O-GDM or O-T1D to the control group. Since we previously observed in adolescent O-GDM that maternal obesity was independently associated with the methylation of *ESM1* and *MS4A3*, we also wanted to assess this relationship in the present cohort. Therefore, we conducted separate multiple regression analyses for each gene in each tissue, including the covariate of maternal pre-pregnancy BMI, with adjustment for offspring sex, as shown in Appendix A. We found no association between methylation levels in blood at any of the examined CpGs, and GDM or maternal pre-pregnancy BMI, when comparing O-BP and O-GDM (Appendix A and Figure 3). Similarly, no association between methylation at the target CpGs and GDM or maternal pre-pregnancy BMI was observed in SAT. 

No differences in DNA methylation between O-T1D and O-BP were detected in blood. We found, however, that in SAT, methylation of *MS4A3* at the position 2 CpG site was 1.8% higher in O-T1D compared to O-BP. We also detected a trend towards a 1.9% higher methylation level in SAT of *MS4A3* (cg14328641), and a 1.5% higher methylation level in SAT of *ESM1* (cg09452568), in O-T1D compared to O-BP (Appendix A and Figure 3).

### 3.3. Gene Expression Profile

First, we examined the relationship between *TSPAN14* and *ESM1* expression in blood and SAT with their corresponding methylation level, to explore whether methylation degree was regulating the expression levels. Neither *TSPAN14* nor *ESM1* were correlated with methylation degree at any of the CpGs (all *p* ≥ 0.53, Figure 4).

Next, we examined whether the expression of the target genes was associated with diabetes in pregnancy and/or maternal pre-pregnancy BMI, utilizing the same approach as described for the methylation data. We used separate multiple regression analyses for each gene in each tissue, to assess the association between maternal obesity and diabetes, with adjustment for offspring sex (Table 2 and Figure 5). No expression changes in the blood were observed between the O-GDM and O-BP. However, in SAT, *ESM1* expression was decreased by 55.4% in O-GDM compared to O-BP (*p* = 0.02), but not associated with maternal pre-pregnancy BMI. Interestingly, SAT *TSPAN14* expression was also negatively associated with GDM (23.2% decreased expression in O-GDM compared to O-BP), and at the same time positively associated with maternal pre-pregnancy BMI (*p* ≤ 0.01), with the offspring presenting 2.6% higher expression per kg/m^2^ increase in BMI (Table 2), corresponding to 20.6% higher levels in expression if the mother had a BMI above 25 kg/m^2^. No changes in *ESM1* or *TSPAN14* expression in either blood or SAT were found in O-T1D compared to O-BP.

### 3.4. Gene Expression Correlations with Clinicals Characteristics

Due to our findings of differential *ESM1* and *TSPAN14* expression in SAT in GDM offspring only, we further investigated whether *ESM1* and *TSPAN14* expression in SAT was associated with clinical parameters of the O-GDM that differed from the O-BP (Table 1), and therefore focused on correlations between gene expression, and offspring diastolic blood pressure and glucose tolerance. Since the O-GDM had higher 2 h glucose levels post OGTT compared to O-BP, we used offspring HbA1c as a clinical parameter for glucose tolerance. Furthermore, because the expression was measured in offspring SAT, we also included total body fat percentage in the correlation analyses, to assess whether the offspring’s own adiposity was potentially affecting the expression levels. We found that *ESM1* expression in SAT was inversely correlated with diastolic blood pressure of the O-GDM group only (*r* = −0.53, *p* = 0.01) (Appendix A, Figure 6). Furthermore, *TSPAN14* expression in SAT was correlated with HbA1c (*r* = 0.29, *p* = 0.03) and body fat percentage (*r* = 0.34, *p* = 0.01) in the O-GDM group only (Appendix A, Figure 6).

## 4. Discussion

Based upon own previous findings in adolescent offspring of GDM women, we investigated the link between maternal diabetes and obesity in pregnancy, and the epigenetic reprogramming of *ESM1*, *MS4A3*, and *TSPAN14* genes, in adult offspring blood cells and subcutaneous adipose tissue. Specifically, we examined whether it was possible to replicate lower DNA methylation degrees at specific genes linked to exposure of maternal obesity and GDM. However, we did not observe any changes in DNA methylation between O-GDM and O-BP at the same CpG sites near the *ESM1*, *TSPAN14*, and *MS4A3* genes that we previously found to have lower methylation degree in blood cells from 9–16 year-old children of women with GDM compared to controls [11]. We also did not observe any association between DNA methylation at these loci and maternal pre-pregnancy BMI, while *ESM1* and *MS4A3* methylation was negatively correlated with maternal pre-pregnancy BMI in our previous study. It is worth noting that we also did not observe any trends in the methylation differences between the offspring groups, as these appeared to go in both directions (both higher and lower) compared to the consistently lower methylation degree that was found in the adolescent O-GDM. It should be noted that the present study cohort of adult offspring was substantially smaller than our discovery study in adolescent offspring; indeed, in the previous study we validated two of the three target genes (*ESM1* and *MS4A3*) in a much larger replication cohort of approximately 900 individuals [11]. Hence, we cannot exclude that the negative validation in the present study to some extent may be due to a lack of statistical power. 

Interestingly, the methylation levels in blood for all three examined genes were lower in the adult offspring (median age 30.5 years) compared to the adolescent offspring from our previous cohort (median age 12 years). Specifically, methylation was ~25% lower in *MS4A3* (from 50% to 22%), ~10% lower in *ESM1* (from 54% to 46% at cg00992687 and from 69% to 63% at cg09452568), and ~8% lower in *TSPAN14* (from 22% to 14%). Notably, it is a limitation that these methylation levels cannot be directly compared, since they were not measured simultaneously; nevertheless, DNA methylation is the most stable and reproducible epigenetic mark known, across methodological approaches [27,28,29]. Therefore, these results might indicate that DNA methylation levels at the candidate genes, at least in blood, are affected by aging. Methylation remodeling due to age may have caused the subtle differences observed in adolescents to disappear in adults [30,31], potentially explaining the absence of methylation changes between O-GDM and O-BP reported here. By extension, we can also not exclude that the missing replication of methylation differences in the adult offspring was caused by differences in the maternal pre-pregnancy BMI of the two separate cohorts, since mothers of the adolescent offspring had a higher average pre-pregnancy BMI of 26.7 kg/m^2^, compared to the older offspring’s mothers with an average pre-pregnancy BMI of 24.3 kg/m^2^.

Finally, we detected an increase in methylation at *MS4A3* in O-T1D in SAT, although not in blood. This observation could, however, likely be due to chance, since we examined and tested multiple CpGs, without correction for multiple testing. Therefore, this finding in O-T1D requires further validation.

To test whether methylation could influence gene expression, we performed a correlation analysis between DNA methylation at target sites and transcript levels, but we did not observe any correlation in either blood or SAT. Moreover, the non-significant difference in methylation of *ESM1* and *TSPAN14* in SAT between O-GDM and O-BP (approximately 0.4–1%) was too small to be relevant for gene expression, which was instead significantly different between these groups. The lack of association between gene expression and methylation indicates that the epigenetic marks studied here are not those primarily responsible for the transcriptional regulation of the selected genes in adult offspring, at least at the time of collection of the tissue samples. In addition, the specific CpG sites investigated were not located in any transcription factor hot spots, but primarily located in exon and intron regions (Figure 2), which may also explain the lack of correlation between methylation and expression, since such associations are more likely to be present in promoter and enhancer regions [32].

The expression of *ESM1* was significantly lower in subcutaneous adipose tissue in GDM, but not T1D offspring. As previously mentioned, *ESM1* encodes for a secreted circulating protein, endocan, which has been implicated in regulation of inflammation and atherosclerosis development [13]. Whether *ESM1* expression in adipose tissue can predict circulating endocan levels is, however, unclear. Janke et al. [33] did not report any correlation between transcript levels in adipose tissue and circulating endocan concentrations. In contrast to our data, this study reported significantly higher adipose *ESM1* RNA expression in obese versus lean women, but lower circulating endocan concentrations among both overweight and obese subjects. The discrepancy with our gene expression results may, however, be due to the difference in cohort size and, thereby, statistical power, since our cohort was larger, or due to the nature of the population studied, since the present cohort of adult offspring consisted of generally healthy, normal weight, and young adults. Although the study participants were all normotensive in the present study, the O-GDM group exhibited higher mean diastolic blood pressure compared to the O-BP group, while the O-T1D blood pressure was similar to the controls. Since *ESM1* expression negatively correlated with diastolic blood pressure in O-GDM, it could be speculated that circulating endocan levels may be partly involved in this association. Systemic endocan concentrations are in fact positively associated with arterial stiffness and atherosclerosis development, and thus endocan has been recognized as a candidate biomarker of cardiovascular health [13]. One study also found significantly higher serum endocan concentrations in T2D patients with subclinical atherosclerosis than in participants with T2D only [34]. Additionally, higher circulating endocan was detected in patients with acute coronary syndrome [35]. Further studies are, therefore, needed to investigate whether GDM and maternal obesity are associated with increased endocan levels in offspring. 

The regression analysis conducted in the present study identified lower expression of *TSPAN14* in SAT from O-GDM compared to controls, but simultaneously *TSPAN14* expression was also independently positively associated with maternal pre-pregnancy BMI. This result suggests that maternal GDM and maternal obesity are likely associated with SAT *TSPAN14* expression through diverse pathways, and that aspects of GDM other than obesity may assert a different influence on offspring gene regulation. The lack of association between maternal T1D and *TSPAN14* expression in the offspring, however, rules out a link between gene regulation and in utero exposure to hyperglycemia alone, since this offspring group was exposed to increased glucose levels from the beginning of pregnancy, and to a higher extent than the offspring of women with GDM, where hyperglycemia generally does not develop before mid-late pregnancy. The factors that in GDM could induce downregulation of *TSPAN14* expression in adipose tissue, and whether the association is due to chance, remain to be confirmed.

To date, a role of *TSPAN14* in metabolic health has, to the best of our knowledge, not been reported. The positive correlations between *TSPAN14* expression levels and total body fat percentage and HbA1c in the offspring led us to hypothesize that this gene may be involved in the metabolic regulation of adipose tissue in O-GDM. The gene product is a member of the TSPANC8 family of proteins, which are mainly known for interacting with the metalloproteinase ADAM10, thereby modulating ADAM10 localization to the membrane and interaction with NOTCH [15,36]. As ADAM10 is implicated in several diseases, including Alzheimer’s dementia, cardiovascular diseases, and inflammation [18], including inflammation of the adipose tissue [37], it is possible that the reprogramming of *TSPAN14* plays a role in cardio-metabolic health by regulating ADAM10 function. Further studies are, therefore, required to examine the functional role of *TSPAN14* in metabolic regulation of the adipose tissue.

For both *ESM1* and *TSPAN14*, O-T1D participants did not exhibit the same expression changes as the O-GDM group. It is noteworthy that the O-T1D group was also characterized by having leaner mothers (Table 1), which suggests that other factors implicated in GDM, such as increased inflammation or lifestyle, and not mere hyperglycemia, may play a role in the epigenetic reprogramming of *ESM1* and *TSPAN14*. We speculate that the combination of maternal hyperglycemia and pre-gestational BMI, or pre-gestational BMI alone, regulate the gene expression changes of *ESM1* and *TSPAN14* in the adipose tissue of GDM offspring. 

It is worth mentioning that in the first follow-up study, where the participants had a mean age of 22 years, the offspring exposed to maternal diabetes had an adjusted risk (odds ratio) of prediabetes and diabetes of 8 in the O-GDM group and of 4 in the O-T1D. A total of 250 participants were lost to follow-up in the present study, and the analysis of their characteristics revealed that the participants who dropped out of the study had significantly higher prevalence of metabolic syndrome, prediabetes, and T2D in the first follow-up [19]. This suggests that the participants in this study were among the healthier offspring of the initial cohort. As a consequence, the findings of the current study might have been more pronounced if it had included offspring who developed metabolic conditions.

## 5. Conclusions

Despite the lack of replication of lower DNA methylation levels at the *ESM1*, *TPSAN14*, and *MS4A3* DNA regions, as previously observed in GDM exposed offspring at childhood age, in this study, we observed differential transcriptional changes of *ESM1* and *TSPAN14* in the adipose tissue of adult offspring exposed to GDM in pregnancy compared to non-exposed offspring. Interestingly, *ESM1* expression was solely linked to GDM status, but *TSPAN14* expression was also linked to maternal pre-pregnancy BMI. *ESM1* and *TSPAN14* have the potential to be involved in cardiometabolic health in GDM offspring, by regulating metabolic pathways, including inflammation, in the adipose tissue. The offspring of mothers with T1D did not show any changes in gene expression, further reinforcing that maternal BMI and other maternal factors, besides maternal hyperglycemia, influence the expression of *ESM1* and *TSPAN14* in offspring.

## Figures and Tables

**Figure 1 biomedicines-10-01244-f001:**
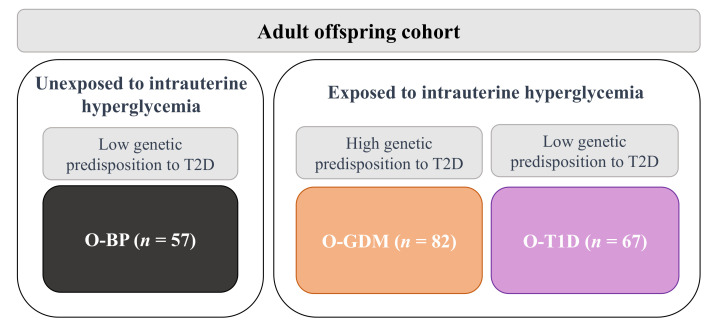
Overview of the adult offspring cohort analyzed in the present study. O-GDM: offspring of women with GDM, O-T1D: offspring of women with T1D, O-BP: offspring of women from the background population, T2D: type 2 diabetes.

**Figure 2 biomedicines-10-01244-f002:**
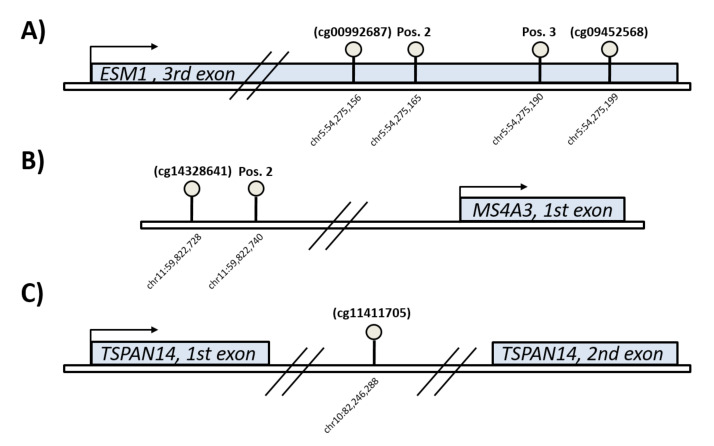
Schematic representation of the genomic location of the target CpGs associated with the genes *ESM1* (Endothelial cell specific molecule 1) (**A**), *MS4A3* (Membrane spanning 4-domains A3) (**B**), and *TSPAN14* (Tetraspanin 14) (**C**). Differentially methylated CpGs detected in the previous cohort of adolescent offspring are shown by their cg number, while the other indicated sites are neighboring CpGs included in the pyrosequencing assay.

**Figure 3 biomedicines-10-01244-f003:**
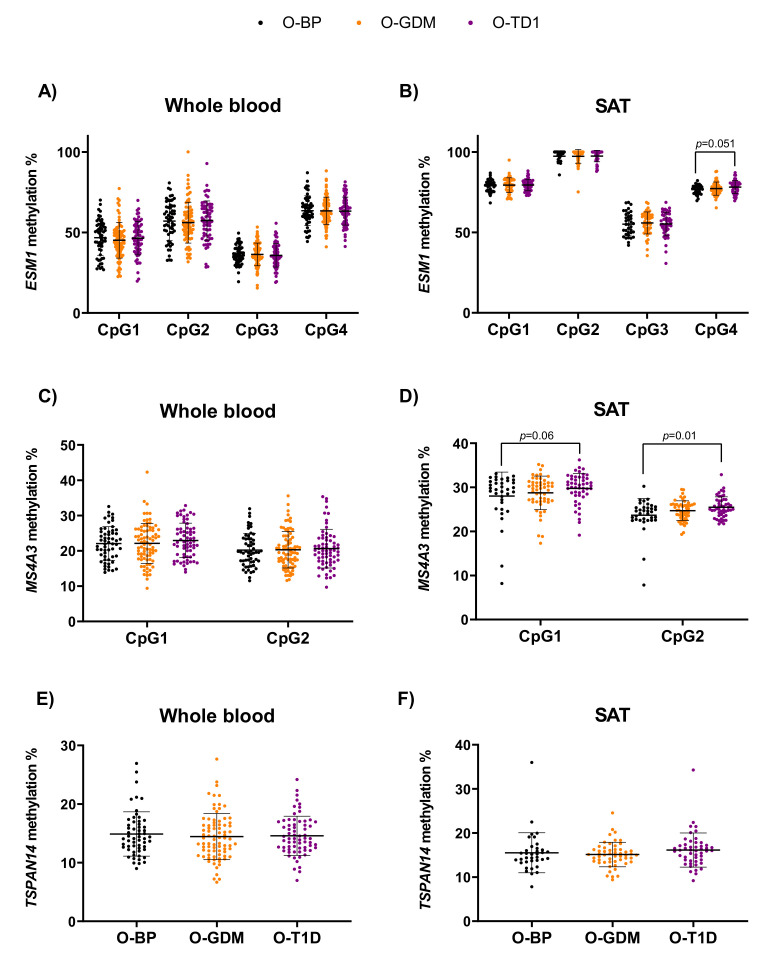
Methylation at target CpG sites associated with *ESM1*, *MS4A3*, and *TSPAN14* in whole blood (**A**,**C**,**E**) and subcutaneous adipose tissue (SAT) (**B**,**D**,**F**). Bars represent mean (± SD). *p*-values represent difference between offspring groups and were calculated by multivariate regression modelling, including the covariates of maternal diabetes status, maternal pre-pregnancy BMI, and offspring sex. Black dots indicate O-BP, orange dots indicate O-GDM, and purple dots indicate O-T1D. O-GDM: offspring of women with GDM, O-T1D: offspring of women with T1D, O-BP: offspring of women from the background population.

**Figure 4 biomedicines-10-01244-f004:**
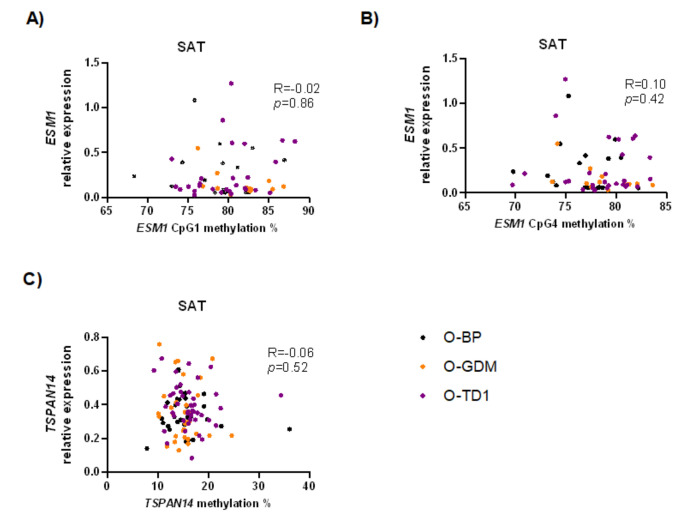
Correlation analysis between subcutaneous adipose tissue (SAT) gene expression and methylation level at target CpG sites of *ESM1* (cg00992687/CpG1 (**A**) and cg09452568/CpG4 (**B**)), and *TSPAN14* (cg11411705 (**C**)). R-values and *p*-values were calculated using a Spearman rank test. Black dots indicate O-BP, orange dots indicate O-GDM, and purple dots indicate O-T1D. O-GDM: offspring of women with GDM, O-T1D: offspring of women with T1D, O-BP: offspring of women from the background population.

**Figure 5 biomedicines-10-01244-f005:**
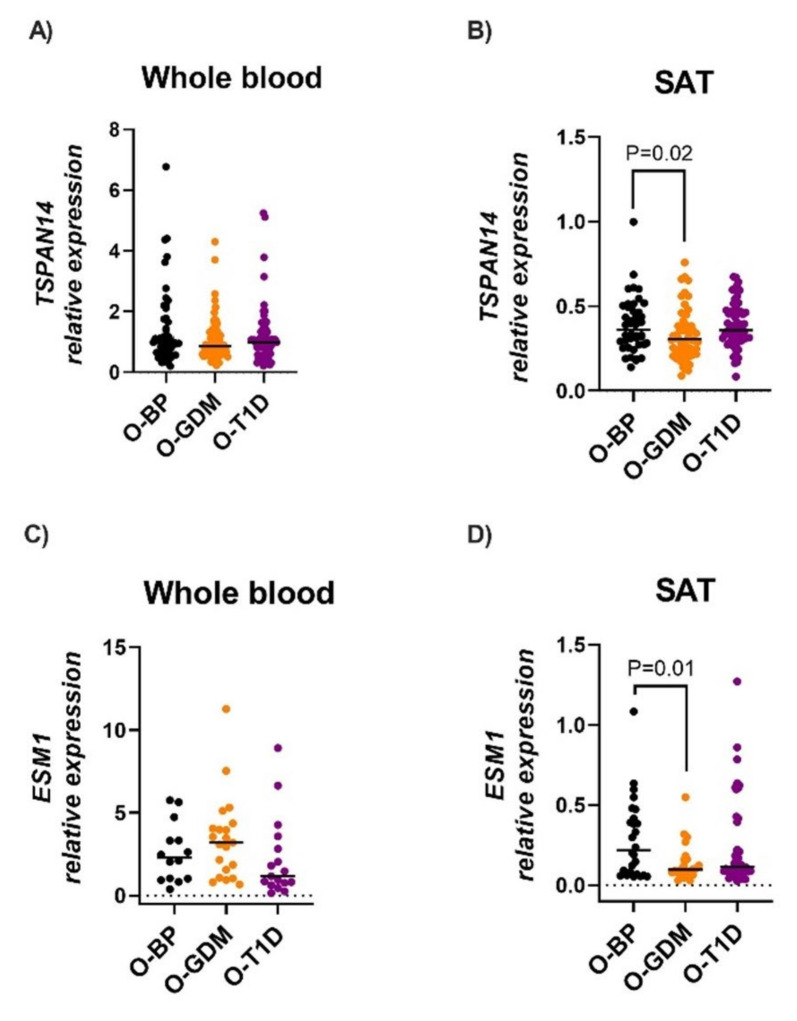
*TSPAN14* (**A**,**B**) and *ESM1* (**C**,**D**) expression in whole blood and subcutaneous adipose tissue (SAT). Bars represent medians. *p*-values represent difference between offspring groups, and were calculated by multivariate regression modelling, including the covariates of maternal diabetes status, maternal pre-pregnancy BMI, and offspring sex. O-GDM: offspring of women with GDM, O-T1D: offspring of women with T1D, O-BP: offspring of women from the background population.

**Figure 6 biomedicines-10-01244-f006:**
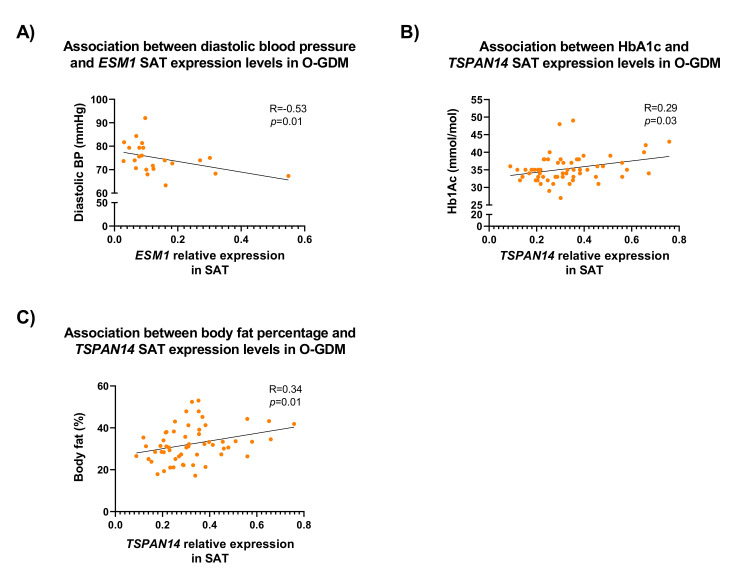
Association between subcutaneous adipose tissue (SAT) *ESM1* and offspring diastolic blood pressure (**A**), and SAT *TSPAN14* expression and offspring HbA1c levels (**B**) and fat percentage (**C**). R-values and *p*-values were calculated using a Spearman rank test. O-GDM: offspring of women with GDM.

**Table 1 biomedicines-10-01244-t001:** Clinical characteristics of the adult offspring cohort, including maternal and birth characteristics.

	O-GDM	O-T1D	O-BP	*p*-Value	
	*n* = 82	*n* = 67	*n* = 57	O-GDM vs. O-BP	O-T1D vs. O-BP
Maternal-BMI (kg/m^2^)	24.3 (±5.6)	21.7 (±1.9)	21.2 (±3.5)	**<0.0001**	0.30
Birth weight (g)	3398 (560)	3322 (719)	3484 (431)	0.33	0.13
Gestational age at birth (days)	274 (269–277)	261 (257–263)	282 (276–287)	**<0.0001**	**<0.0001**
Large for gestational age (yes vs. no)	17% (14/82)	42% (28/67)	12% (7/57)	0.44	**<0.0001**
Gender (female)	39 (47.6)	36 (53.7)	31 (54.4)	0.43	0.94
Age (years)	30.8 (±2.1)	31.3 (±2.4)	31.3 (±2.4)	0.27	0.83
BMI (kg/m^2^)	24.7 (21.8–27.1)	24.2 (22.1–27.7)	24.2 (21.7–26.6)	0.71	0.46
Total body fat (%)	31.2 (±9.1)	32.5 (±9.8)	29.8 (±7.9)	0.35	0.09
Systolic BP (mmHg)	116.8 (±9.1)	116.7 (±8.8)	115.8 (±11.9)	0.61	0.63
Diastolic BP (mmHg)	73.5 (±7.4)	70.8 (±9.0)	70.5 (±7.3)	**0.02**	0.86
HbA1c (mmol/mol)	35 (33–37)	35 (32–36.5)	34 (33–36)	0.10	0.30
Fasting glucose (mmol/L)	4.9 (4.5–5.1)	4.9 (4.7–5.2)	4.9 (4.6–5.1)	0.90	0.59
2 h glucose (mmol/L)	6.0 (±1.81)	6.3 (±1.69)	5.3 (±1.23)	**0.02**	**0.001**

Data are presented as mean (±SD), median (IQR), or counts (percentage). O-GDM: offspring of women with GDM, O-T1D: offspring of women with T1D, O-BP: offspring of women from the background population, Maternal-BMI: maternal pre-gestational BMI, BP: blood pressure. The differences between O-GDM vs. O-BP or O-T1D vs. O-BP for the continuous variables with parametric and non-parametric distributions were tested using an independent samples *t*-test or Mann–Whitney U test, respectively. For the categorical variable (gender) the distribution differences were assessed by the χ^2^ test. *p* values < 0.05 are in bold.

**Table 2 biomedicines-10-01244-t002:** Association between gene expression of *ESM1* and *TSPAN14* and maternal diabetes in pregnancy and pre-pregnancy BMI.

O-GDM Compared to O-BP:						
	Association to Group (O-GDM vs. O-BP)			Association to Maternal Pre-Pregnancy BMI		
Gene	β (95% CI)	% difference in expression	*p*-value	β (95% CI)	% difference in expression	*p*-value
Blood mRNA expression (*n* = 35–126)						
*ESM1*	0.8 (−1.0, 2.6)	30.9%	*0.38*	−0.02 (−0.2, 0.2)	−0.8%	*0.81*
*TSPAN14*	−0.3 (−0.7,0.05)	−21.5%	*0.09*	−0.01 (−0.04, 0.03)	−0.7%	*0.70*
SAT mRNA expression (*n* = 49–100)						
*ESM1*	−0.16 (−0.3, −0.03)	**−55.4%**	** *0.02* **	0.002 (−0.1, 0.1)	0.7%	*0.71*
*TSPAN14*	−0.09 (−0.15, −0.02)	**−23.2%**	** *0.01* **	0.01 (0.001, 0.1)	**2.6%**	** *0.02* **
**O-T1D compared to O-BP:**						
	Association to Group (O-T1D vs. O-BP)			Association to maternal pre-pregnancy BMI		
	β (95% CI)	% difference in expression	*p*-value	β (95% CI)	% difference in expression	*p*-value
Blood mRNA expression (*n* = 31–115)						
*ESM1*	−0.48 (−2.1, 1.1)	−18.6%	*0.55*	−0.22 (−0.5, 0.1)	−8.5%	*0.14*
*TSPAN14*	−0.23 (−0.7, 0.2)	−16.5%	*0.29*	−0.03 (−0.1, 0.1)	−2.2%	*0.52*
SAT mRNA expression (*n* = 68–82)						
*ESM1*	−0.04 (−0.2, 0.1)	−10.3%	*0.61*	0.001 (−0.02, 0.03)	0.3%	*0.97*
*TSPAN14*	−0.01 (−0.1, 0.1)	−2.6%	*0.79*	0.01 (−0.01, 0.02)	2.6%	*0.17*

Estimated difference in relative gene expression degree between O-BP and O-GDM/O-T1D is presented as β (95% CI) and *p*-value, as calculated in the regression models, including the covariates of maternal diabetes status, pre-pregnancy maternal BMI (continuous variable, per BMI unit), and offspring sex. O-GDM: offspring of women with GDM, O-T1D: offspring of women with T1D, O-BP: offspring of women from the background population, SAT: subcutaneous adipose tissue, mBMI: maternal pre-gestational BMI. *p*-values < 0.05 are presented in bold.

## Data Availability

Data is contained within the article and Supplementary Material, and can be accessed from the corresponding author on reasonable request.

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
