# Peer review of "DNA Methylation and Gene Expression in Blood and Adipose Tissue of Adult Offspring of Women with Diabetes in Pregnancy—A Validation Study of DNA Methylation Changes Identified in Adolescent Offspring"

_biomedicines, 2022, doi:10.3390/biomedicines10061244_

Round 1
Reviewer 1 Report
In this article the authors measured DNA methylation from previously identified CpG sites in three genes (ESM1, MS4A3 and TSPAN14) from blood and subcutaneous adipose tissue from an adult offspring cohort exposed to maternal gestational diabetes or Type-1 diabetes in utero, compared to adult offspring from background population. Gene expression was also measured. Overall, results from a previous study (using a different cohort of younger (adolescent) offspring exposed to maternal diabetes in utero) were not replicated, however this study was not a follow-up from the previous study, the new cohort was composed of different, older individuals of about 30 years old (in average), and epigenetic marks (such as 5-methylcytosine) are dynamic and change with age and lifestyle (among other factors).
I suggest the authors, if possible, provide all measurements per patient (pregnant women) including glucose levels during pregnancy in a supplementary Table, this would be important to verify hyperglycemia in pregnancy. This is important because the authors assume that T1D offspring were exposed to higher levels of glucose, however T1D women were treated (and supposedly controlled) with insulin from the time of diagnosis through life, glucose levels in these patients should have been close to the levels found in the background population (unless they had complications). On the other hand, GDM women experienced a sudden increase of glucose levels around the third trimester of pregnancy, exposing the fetus to a hyperglycemic environment until diagnose and dietary intervention, and even with dietary intervention the uterus could still be hyperglycemic in some patients; other factors from the mother (age, genetic predisposition, etc) may also be relevant.
Methylation in CpG sites usually affects gene expression if methylated cytosines are located within regulatory sequences (e.g. transcription factor binding sites) in the promoter or in the 5’-UTR region. In this study, CpG sites were located within the third exon from the ESM1 gene and in an intron from the TSPAN14 gene. Also only one CpG site was analyzed from the MS4A3 promoter. I understand the authors were comparing these results with a previous study; however promoter methylation usually provides more information and correlates better with gene expression; probably the authors would analyze other CpG sites within the promoter region of each gene?
Please indicate what SAT biopsy stands for (abdominal subcutaneous adipose tissue?)
Indicate how genomic DNA was removed from total RNA, and why was TFIIB selected as reference gene, was this gene tested for stability? This is important because both complete genomic DNA removal from total RNA and reference gene stability are absolutely necessary to obtain reliable results. Also, indicate the method used to calculate relative gene expression.
Lines 209-211: The authors stated “Both groups exposed to diabetes in pregnancy presented with higher 2hr plasma glucose concentrations during the OGTT when compared to O-BP (p≤0_.001)…”, however glucose values were within normal range (according to values reported by WHO).
Figure S1 should be included in the main text. Did the authors perform correlation analysis for GDM, T1D, and BP separately?
Author Response
In this article the authors measured DNA methylation from previously identified CpG sites in three genes (ESM1, MS4A3 and TSPAN14) from blood and subcutaneous adipose tissue from an adult offspring cohort exposed to maternal gestational diabetes or Type-1 diabetes in utero, compared to
adult offspring from background population. Gene expression was also measured. Overall, results from a previous study (using a different cohort of younger (adolescent) offspring exposed to maternal
diabetes in utero) were not replicated, however this study was not a follow-up from the previous study, the new cohort was composed of different, older individuals of about 30 years old (in average), and epigenetic marks (such as 5-methylcytosine) are dynamic and change with age and lifestyle (among
other factors).
Comment 1
I suggest the authors, if possible, provide all measurements per patient (pregnant women) including glucose levels during pregnancy in a supplementary Table, this would be important to verify hyperglycemia in pregnancy. This is important because the authors assume that T1D offspring were exposed to higher levels of glucose, however T1D women were treated (and supposedly controlled) with insulin from the time of diagnosis through life, glucose levels in these patients should have been close to the levels found in the background population (unless they had complications). On the other
hand, GDM women experienced a sudden increase of glucose levels around the third trimester of pregnancy, exposing the fetus to a hyperglycemic environment until diagnose and dietary intervention, and even with dietary intervention the uterus could still be hyperglycemic in some patients; other factors from the mother (age, genetic predisposition, etc) may also be relevant.
Author response:
Regarding the difference in severity of maternal hyperglycemia in women with GDM and type 1 diabetes we appreciate your suggestion to provide more details on the pregnant women´s clinical
characteristics, especially glucose levels during pregnancy, in a supplementary. This we have now provided, and it is shown in Table S2.
It is known from newer studies performing measurements of consecutive HbA1c-values during pregnancy, that women with type 1 diabetes experiences higher levels of glucose during pregnancy, than women with diet-treated GDM. Unfortunately, HbA1c was not introduced to clinical practice in
the period 1978-85, wherefrom the participants in our study were born. Thus, in the supplementary table, we present the available glucose values during pregnancy, which were less accurate surrogate
measures of glycemia (from the diagnostic OGTT and in-hospital 3-day 7-point profiles, from women with GDM and type 1 diabetes, respectively). These measures are not suitable to compare levels of glycemia between type 1 diabetes and GDM pregnancies, since they were obtained at different
timepoints, and by different methods, but still enables us to some extent to evaluate the effect of different glycemic levels within the two types of pregnancy – and to evaluate the association between maternal glucose measures and offspring outcomes.Nevertheless, even today where women with T1D has access to much better medications, insulin pumps, home glucose monitoring, etc., the glucose levels in pregnant women with type 1 diabetes
fluctuates substantially more and the HbA1c level is higher than seen in women with diet-treated GDM at least in the first part of pregnancy. This is further supported by the higher rate of newborns being large of gestational age among women with type 1 diabetes.
The following sentence is attempting to modify our statement regarding differences in severity between glycemia in diet treated GDM and type 1 diabetes:
Line 208-213:
“The clinical characteristics of the pregnant women are summarized in Table S2. Glucose levels during pregnancy were obtained from women with T1D and GDM at different time points. The degree of intrauterine hyperglycemia was most likely of higher severity in women with overt T1D, versus GDM as only women with mild GDM (diet-treated) were included in the study. This assumption is
further supported by the markedly higher rate of LGA infants among O-T1D.”
Comment 2
Methylation in CpG sites usually affects gene expression if methylated cytosines are located within regulatory sequences (e.g. transcription factor binding sites) in the promoter or in the 5’-UTR region.
In this study, CpG sites were located within the third exon from the ESM1 gene and in an intron from the TSPAN14 gene. Also, only one CpG site was analyzed from the MS4A3 promoter. I understand the authors were comparing these results with a previous study; however, promoter methylation usually
provides more information and correlates better with gene expression; probably the authors would analyze other CpG sites within the promoter region of each gene?
Author response:
We fully agree with the observation that methylation at the CpG sites investigated in the current study
might not be relevant in gene expression regulation, as methylation at only single CpGs was measured, and some of the sites are not located within standard regulatory regions (i.e. the gene promoter). It is
known, however, that methylation within gene bodies might also correlate with gene expression.
Specifically, methylation within the first intron (as for TSPAN14) is also negatively associated with gene expression (reference: Consistent inverse correlation between DNA methylation of the first intron and gene expression across tissues and species –PubMed (nih.gov)), while methylation in gene bodies
has, a least in some cases, a positive correlation with gene expression (references: Replication timingrelated and gene body-specific methylation of active human genes. –PubMed (nih.gov), Targeted and genome-scale strategies reveal gene-body methylation signatures in human cells – PubMed (nih.gov)).
Consequently, investigating whether differentially methylated sites detected in epigenome wide studies are actually relevant in regulating gene expression, despite their location, might be of interest.
We agree that analyzing methylation at a larger region, and potentially at the promoter, will be a very interesting next step in future research of these candidate genes to be able to examine whether this
epigenetic mark plays an active role in regulating the target genes, however it is outside of the scope of the current work.
Comment 3
Please indicate what SAT biopsy stands for (abdominal subcutaneous adipose tissue?)
Author response:
Thank you for this comment. We have added the information of “abdominal” subcutaneous adipose tissue in the manuscript, where it is first described.
Line 134:
“Abdominal SAT biopsy samples were collected after overnight fasting.”
Comment 4
Indicate how genomic DNA was removed from total RNA, and why was TFIIB selected as reference gene, was this gene tested for stability? This is important because both complete genomic DNA
removal from total RNA and reference gene stability are absolutely necessary to obtain reliable results.
Also, indicate the method used to calculate relative gene expression.
Author response:
Genomic DNA was removed from total RNA by DNase I treatment during RNA purification. This have now been added in the methods section:
Line 173-174:
“Genomic DNA was removed from total RNA by DNase I treatment during RNA extraction.”
Thank you for the questions regarding the choice of reference gene. TFIIB was chosen as reference gene based on its known capacities of not changing with adipogenesis nor by altered glucose levels
(references: Genome-wide profiling of PPARγ:RXR and RNA polymerase II occupancy reveals temporal activation of distinct metabolic pathways and changes in RXR dimer composition during adipogenesis - PMC (nih.gov), Integrative Genomics Outlines a Biphasic Glucose Response and a
ChREBP-RORγ Axis Regulating Proliferation in β Cells - PubMed (nih.gov)).
We tested the reference gene by establishing a cDNA pool from all samples, and we conducted a relative quantification of expression based on a standard curve derived from the pooled cDNA.
We have added this important information in the methods section, as shown below:
Line 180-182:
“TFIIB was chosen as reference gene based on its known capacities of not changing with adipogenesis nor by altered glucose levels [23,24].”
Comment 5
Lines 209-211: The authors stated “Both groups exposed to diabetes in pregnancy presented with higher 2hr plasma glucose concentrations during the OGTT when compared to O-BP (p≤0_.001)…”, however glucose values were within normal range (according to values reported by WHO).
Author response:
Yes, it is correct that both groups are within the normal range. In this sentence we are showing that the two hyperglycemia exposed offspring groups have significantly higher glucose levels post the OGTT, when compared to the offspring of the background population.
We have now clarified this as shown below:
Line 217-220:
“Both groups exposed to diabetes in pregnancy presented with higher 2hr plasma glucose concentrations during the OGTT when compared to O-BP (p≤0.001), however, the glucose levels of all three offspring groups were still within the normal range.”
Comment 6
Figure S1 should be included in the main text. Did the authors perform correlation analysis for GDM, T1D, and BP separately?
Author response:
We thank the reviewer for the suggestion. We have now moved the suppl. figure into the main manuscript, and it is now entitled Figure 4.
Correlation between DNA methylation at target sites and gene expression shown in Figure S1 was performed combining all three groups. Even when the analysis was performed separately for each group, no correlation was observed.
Reviewer 2 Report
The paper by Manitta et al is well-structured and easy to follow. I only have a few comments and questions about the statistics and methylation changes.
In Table S2 the authors present the association between DNA methylation of ESM1, TSPAN14 and MS4A3, and maternal diabetes in pregnancy and pre-pregnancy BMI. They use a p-value < 0.05 as the significance level. It is not clear to me from the description of the statistical analysis if they adjust for multiple comparisons. Is it appropriate with some kind of adjustment? This question is also relevant to Table 2 (and maybe other tables/graphs?).
Methylation levels for the examined genes were lower in blood in the adult offspring compared to the adolescent offspring. Did you see the same trend in SAT? How large was the % point change in the adult offspring cohort compared to the adolescent offspring? In figure 3, the methylation % point change for ESM1 and TSPAN14 in SAT seems to be around one. Do you expect this to be sufficient to cause changes in gene expression?
Author Response
Comment 1
In Table S2 the authors present the association between DNA methylation of ESM1, TSPAN14 and MS4A3, and maternal diabetes in pregnancy and pre-pregnancy BMI. They use a p-value < 0.05 as the significance level. It is not clear to me from the description of the statistical analysis if they adjust
for multiple comparisons. Is it appropriate with some kind of adjustment? This question is also relevant to Table 2 (and maybe other tables/graphs?).
Author response:
We fully agree with the reviewer regarding the analyses presented in table S2, they are likely affected
by multiple testing. Still, these results are mainly negative, since only two CpGs are shown to be changed in the T1D offspring. Due to the mostly negative results, we instead of conducting the
adjustment for multiple testing, chose to write in the discussion that this finding is likely caused by multiple testing, and hence a false positive. We have now clarified this further in the discussion as shown below:
Line 344-34:
“Finally, we did detect an increase in methylation at MS4A3 in O-T1D in SAT, although not in blood.
This observation could, however, likely be due to chance, since we have examined and tested multiple CpGs, without correction for multiple testing. Therefore this finding in O-T1D requires further
validation.”
Regarding the same question for Table 2, we do not think that correcting for multiple testing is appropriate here, since only four statistical tests (linear regression models) are conducted, and each was based on a primary hypothesis.
Comment 2
Methylation levels for the examined genes were lower in blood in the adult offspring compared to the adolescent offspring. Did you see the same trend in SAT? How large was the % point change in the
adult offspring cohort compared to the adolescent offspring? In figure 3, the methylation % point change for ESM1 and TSPAN14 in SAT seems to be around one. Do you expect this to be sufficient to cause changes in gene expression?
Author response:
We thank the reviewer for the question. SAT methylation was not analyzed in the previous cohort of adolescent offspring and we were therefore unable to compare methylation levels between the two cohorts in this tissue. As mentioned in the discussion, methylation was approximately 25% lower at
MS4A3, 7-8% lower at ESM1 and 8% lower at TSPAN14, comparing older and younger offspring. We agree with the reviewer that the methylation change was very small (~1%) at the examined regions
and not significant for O-GDM SAT. Although small changes in methylation might have a bigger effect on gene expression, we do not expect a 1% difference in methylation levels to impact gene expression. We therefore believe that other mechanisms, or methylation at other regions, are responsible for the observed change in gene expression of TSPAN14 and ESM1.
We have now specified this point in the discussion as shown below:
Line 350-353:
“Moreover, the non-significant difference in methylation of ESM1 and TSPAN14 in SAT between O-GDM and O-BP (approximately 0.4-1%) is too small to be expected as relevant for gene expression, which was instead
significantly different between these groups.
Round 2
Reviewer 1 Report
In this revised version of the manuscript the authors answered all my questions, I am quite satisfied with the answers, and the effort is appreciated. I have no further comments.